# Preparation and Antibacterial Activity of Thermo-Responsive Nanohydrogels from Qiai Essential Oil and Pluronic F108

**DOI:** 10.3390/molecules26195771

**Published:** 2021-09-23

**Authors:** Jianfeng Zhan, Feng He, Shuxian Chen, Abishek Jung Poudel, Ying Yang, Lin Xiao, Fu Xiang, Shiming Li

**Affiliations:** 1Hubei Key Laboratory of Economic Forest Germplasm Improvement and Resources Comprehensive Utilization, Huanggang Normal University, Huanggang 438000, China; zhanjianfeng2010@163.com (J.Z.); suzyxian@hnu.edu.cn (S.C.); 2Hubei Collaborative Innovation Center for the Characteristic Resources Exploitation of Dabie Mountains, Huanggang Normal University, Huanggang 438000, China; 3Department of Biomedical Engineering, College of Life Science and Technology, Huazhong University of Science and Technology, Wuhan 430074, China; abishekpoudel2025@hotmail.com (A.J.P.); xiaol@hust.edu.cn (L.X.); 4Institute of Resource Biology and Biotechnology, Department of Biotechnology, College of Life Science and Technology, Huazhong University of Science and Technology, Wuhan 430074, China; yangying@hust.edu.cn; 5Department of Food Science, Rutgers University, New Brunswick, NJ 08901, USA

**Keywords:** essential oil, qiai, nanohydrogel, thermo-responsive property, antibacterial activity

## Abstract

Essential oils (EOs) have been used in cosmetics and food due to their antimicrobial and antiviral effects. However, the applications of EOs are compromised because of their poor aqueous solubility and high volatility. Qiai (*Artemisia argyi* Levl. et Van. var. *argyi* cv. Qiai) is a traditional Chinese herb and possesses strong antibacterial activity. Herein, we report an innovative formulation of EO as nanohydrogels, which were prepared through co-assembly of Qiai EO (QEO) and Pluronic F108 (PEG-*b*-PPG-*b*-PEG, or PF108) in aqueous solution. QEO was efficiently loaded in the PF108 micelles and formed nanohydrogels by heating the QEO/PF108 mixture solution to 37 °C, by the innate thermo-responsive property of PF108. The encapsulation efficiency and loading capacity of QEO reached 80.2% and 6.8%, respectively. QEO nanohydrogels were more stable than the free QEO with respect to volatilization. Sustained QEO release was achieved at body temperature using the QEO nanohydrogels, with the cumulative release rate reaching 95% in 35 h. In vitro antibacterial test indicated that the QEO nanohydrogels showed stronger antimicrobial activity against *S. aureus* and *E. coli* than the free QEO due to the enhanced stability and sustained-release characteristics. It has been attested that thermo-responsive QEO nanohydrogels have good potential as antibacterial cosmetics.

## 1. Introduction

Essential oils (EOs), also known as aromatic or volatile oils, are oily liquids extracted from different parts of aromatic plants by steam distillation, solvent extraction or dry distillation. EOs are mainly composed of monoterpenes, sesquiterpenes, aromatic alcohols and sesquiterpene hydrocarbons. In recent years, EOs have been found to show extensive biological activities, such as antidepressive [1], anti-inflammatory [2], anti-oxidant [3], antiviral [4] and antimicrobial activity [5,6,7]. However, the application of EOs is limited due to their volatility and poor water solubility. Currently, EOs are usually utilized as compounds with other pure oils for cosmetic and therapeutic applications, which may have low bioavailability and high cost.

In the past decade, researchers have explored the application of nanotechnology in formulating EOs. EOs were loaded in different nanocarrier systems such as nanoparticles [8,9] nanocapsules [10,11], and nanofibre membranes [12,13], or processed to nanoemulsions [14]. These methods have addressed the issue of water insolubility of EOs. However, some other problems, such as reduced EO loading capacity, low encapsulation efficiency, poor stability, and biocompatibility of the materials, may remain or emerge.

Poly(ethylene glycol)-*block*-poly(propylene glycol)-*block*-poly(ethylene glycol) (PEG-*b*-PPG-*b*-PEG, i.e., Pluronic F108, denoted as PF108), a difunctional block copolymer, is employed in drug delivery systems and cell culture due to its low cytotoxicity and innate thermo-responsive property [15,16,17]. The thermo-responsive property of PF108 is attributed to the PPG block, which is water-soluble at low temperatures, while it becomes insoluble at higher temperatures [18,19]. At low temperatures and low concentrations, PF108 can be dissolved in water as free molecular chains. When increasing the temperature, PF108 molecules tend to assemble into micelles, with the PPG segments forming the cores. Furthermore, gelation may occur through self-crosslinking of the micelles at high polymer concentrations. The thermo-responsive micelles and hydrogels based on PF108 have been used as efficient platforms as carriers of drugs and cells for biomedical applications [15,16,17].

Qiai essential oil (QEO) was mainly composed of thujone (14.94%), globulol (11.6%), 1,8-cineole (11.31%), 1-caryophyllene (8.11%), camphor (6.87%), α-terpineol, (4.69%), caryophyllene oxide (4.05%), borneol (3.95%), eugenol (3.01%), and germacrene D (2.81%), according to our measurement with gas chromatography-mass spectrometry. The antimicrobial mechanism of QEO was allegedly involved in simultaneous disruption of cell structures, such as the cytomembrane, which led to the leakage of essential molecules, such as protein and K^+^, resulting in cell death [20].

In the present study, PF108 was used for the encapsulation of QEO to develop an innovative QEO formulation as nanohydrogels. The size and morphology of the PF108 micelles, induced by high temperatures (such as body temperature), were determined by dynamic light scattering (DLS) and transmission electron microscopy (TEM), respectively. The QEO nanohydrogels were characterized by Fourier transform infrared spectrum (FTIR) and X-ray diffraction (XRD). The encapsulation efficiency, loading capacity of QEO were then evaluated followed by the study of QEO release in vitro. Finally, the antibacterial activities of QEO nanohydrogels were investigated. The thermo-responsive property of the QEO nanohydrogels could potentially allow them to be used as antibacterial gel cosmetics at body temperature, and convert to liquid status at low temperatures for storage and transportation.

## 2. Results and Discussion

### 2.1. Morphology and Particle Size of QEO-Loaded PF108 Micelles

TEM was used to determine the morphology and particle size of the PF108 micelles and QEO-loaded micelles. The blank micelles (Figure 1A) showed a spherical morphology with a diameter ranging from 100 to 160 nm, while the QEO-loaded micelles (Figure 1B) were observed to show a shrunken and deformed spherical morphology with the particle size ranging from 80 to 100 nm. The morphological and size changes of the QEO-loaded micelles compared with the blank micelles might be due to the evaporation of QEO during sample drying. Liakos et al. also found that cellulose acetate nanocapsules were almost twice the size of cellulose acetate/lemongrass EO nanocapsules, according to TEM results [21]. The particle size of micelles was further studied by DLS. As shown in Figure 2A, the majority (90%) of the blank micelles showed a particle size ranging from 120 to 200 nm, with a very small portion (10%) of the micelles ranging from 45 to 100 nm. The size decrease was also verified by DLS for the QEO-loaded micelles. As shown in Figure 2B, 85% of the QEO-loaded micelles possessed a size varying from 110 to 180 nm, while the others were between 10 and 50 nm. These results are well accordant with the TEM observations. Due to the advantages of nanocarriers such as high specific surface areas, the EO-loaded micelles may lead to high encapsulation efficiency and sustained release of EO [22,23,24,25].

### 2.2. Determination of PF108 Concentration for EO Nanohydrogels by Rheological Method

The temperature dependence of elastic modulus (G’) and viscous modulus (G”) of the blank PF108 hydrogels was studied using the rheological measurement. As shown in Figure 3, the lower critical solution temperature (LCST, i.e., the gelation temperature of the blank PF108 hydrogels) indicated by the crossover point of G’ and G”, decreased as the polymer concentration increased. The LCST values were 34, 32, and 28 °C for the hydrogels of 15%, 20%, and 25% (*w*/*v*) PF108 polymer, respectively. Besides the gelation temperature, it is worth noting that the G’ at 37 °C is also an important parameter for the hydrogels to be used as cosmetics. It can be seen in Figure 3 that the G’ at 37 °C were 206, 930, and 6280 Pa for the hydrogels of 15%, 20%, and 25% (*w*/*v*) PF108 polymer, respectively. These rheological studies suggested that the concentration of the PF108 polymer played a vital role in the formation and strength of the hydrogels. As the concentration of the polymer increases, the micelles are sufficiently entangled to promote gel formation [26]. It is well known that a hydrogel cosmetics product is supposed to have a gelation temperature close to the body temperature as well as a sufficient and comforting strength [27]. In this regard, the hydrogel of 20% (*w*/*v*) PF108 polymer was selected as the EO-loading matrix in the subsequent studies.

### 2.3. Properties of EO Nanohydrogels 

The FTIR spectra of QEO, PF108 hydrogels, and QEO nanohydrogels are presented in Figure 4. The QEO showed characteristic peaks at 3465 cm^−1^ (OH stretching), 2927 cm^−1^ (C-H stretching), 1743 cm^−1^ (C=O stretching), and 1457 cm^−1^ (CH_2_ bending) (Figure 4A) [28]. Peaks at 2888 cm^−1^ (C-H bending), 1467 cm^−1^ (CH_2_ bending), and 1108 cm^−1^ (C-O-C stretching) were observed in the PF108 hydrogels (Figure 4B). Characteristic peaks of QEO and the PF108 hydrogels were typically found in the spectra of QEO nanohydrogels (Figure 4C), which indicated no chemical interaction between the QEO and hydrogel matrix. Moreover, the peaks in the range of 500–1500 cm^−1^ of the QEO nanohydrogels were relatively increased in comparison with the spectra of QEO and the PF108 hydrogels. The results indicated that QEO might be encapsulated into the PF108 hydrogels.

The crystallographic structures of PF108 powder, PF108 hydrogels, and QEO nanohydrogels were determined by XRD (Figure 5). Characteristic peaks at 2θ = 15.4–28.32° were observed, which indicated that all three samples had a certain degree of crystallization. Moreover, the PF108 powder and PF108 hydrogels were observed to have almost the same spectrum, showing the characteristic peaks for the PEG crystalline phase at 2θ values of 19.68° and 22.94° (Figure 5A,B), which is similar to the XRD results of PEG reported by Ahmad [29]. This indicated that the entanglement between micelles did not affect the crystal structure of the copolymer. Compared with the PF108 powder and PF108 hydrogels, the peak at 2θ of 22.94° in the QEO nanohydrogels was enhanced, which implied that the incorporation of QEO led to a crystalline structural change in PF108 (Figure 5C).

### 2.4. Determination of Encapsulation Efficiency and Loading Capacity

The EE% and LC% of QEO in the QEO nanohydrogels were calculated to be 80.2% and 6.8%, respectively. This represents a rather high QEO-loading ability achieved by nanocarriers. As a comparison, the EE% and LC% of alginate/cashew gum nanoparticles loaded with *Lippia sidoides* EO were 55% and 4.4%, respectively [30]. Meanwhile, the EE% of chitosan nanoparticles loaded with *Carum copticum* EO were 26.9% [31]. The higher loading ability of QEO in EO-nanohydrogels might be attributed to the dual mechanisms of hydrophobic interactions in the micellar core and the entanglement between micellar coronas, which effectively locked the QEO within the hydrogel.

### 2.5. In Vitro Release Studies

The QEO release profiles at 37 °C are illustrated in Figure 6. It can be seen that the pure QEO vaporized very rapidly at 37 °C and evaporated completely at 20 h. On the contrary, QEO was released from the QEO nanohydrogels much more slowly. The cumulative release rate of QEO was ca. 78% at 20 h. The sustained QEO release lasted for 35 h, with a final cumulative release rate of ca. 95%. This indicated that the QEO nanohydrogels could achieve a more sustained QEO release than the free QEO at body temperature, which is favorable for the cosmetics application.

### 2.6. Antibacterial Activities of QEO Nanohydrogels

The antibacterial efficacy of QEO nanohydrogels was tested against Gram-positive *S. aureus* and Gram-negative E. coli. As shown in Figure 7, n-hexane and the blank PF108 hydrogels showed no antimicrobial activity against *S. aureus* and *E. coli*. In contrary, both QEO and QEO nanohydrogels showed strong antimicrobial activities. QEO showed stronger antimicrobial activity against both *S. aureus* and *E. coli* than QEO nanohydrogels at 12 h (*p* < 0.05) because of its faster diffusion. However, the antimicrobial activities of QEO and QEO nanohydrogels observed at 24 h were comparable against both *S. aureus* and *E. coli*. It is noted that QEO nanohydrogels showed stronger antimicrobial activity against both *S. aureus* and *E. coli* than QEO at 36 h (*p* < 0.05). This indicated that QEO nanohydrogels have more lasting antibacterial efficacy as compared with free QEO, which is consistent with the in vitro QEO release results. The mycelial growth of A. niger could not be completely inhibited by free clove essential oil (CEO) at a concentration as high as 3 mg/mL, while CEO encapsulated by chitosan nanoparticles could completely inhibit the fungal growth at 1.5 mg/mL [32]. Lemon myrtle essential oil (LM-EO) showed enhanced antibacterial activity against S. *aureus*, *L. monocytogenes*, and *E.coli* compared to LM-EO alone [33]. Similar results were achieved by nanoencaged EO than free EO because of the reduced EO volatility. The superior antibacterial activity, together with the good biocompatibility of PF108, may lead to a bright prospect of EO nanohydrogels as antibacterial cosmetics.

## 3. Materials and Methods

### 3.1. Materials and Chemicals

Pluronic F108 (PEG_133_-PPG_50_-PEG_133_, Mn = 14,600) was purchased from Aladdin Reagent Co., Ltd. (Shanghai, China). Qiai essential oil (purity: 99.6%) obtained from *Artemisia argyi* Levl. et Van. var. *argyi* cv. Qiai, a geographical indication plant of China, was provided by Hubei Lishizhen Biotechnology Co., Ltd. (Huanggang, China). *Staphylococcus aureus* ATCC 29213 (*S. aureus*) and *Escherichia coli* ATCC 25922 (*E. coli*) were provided from Huanggang Normal University, Huanggang, China. Other analytical reagents were purchased from Sinopharm Chemical Reagent Co., Ltd. (Shanghai, China).

### 3.2. Preparation and Characterization of QEO-Loaded PF108 Micelles

QEO-loaded PF108 micelles were prepared following the method adapted from literature [34]. Briefly, 100 mg of PF108 were dissolved in 10 mL of deionized water at 4 °C. QEO (50 μL) was then added to the PF108 solution with rigorous stirring to produce a white mixture solution. The resulted solution was heated at 37 °C for 10 min to obtain the QEO-loaded micellar solution. The blank PF108 micelles were also prepared by the same method except the absence of QEO. 

The particle size of the QEO-loaded micelles and blank micelles were determined using a dynamic light scattering (DLS) instrument equipped with a 4 mW He–Ne laser source (Malvern Instruments, Ltd., Malvern, UK). The micellar solutions were diluted to 0.05% (*w*/*v*) and filtered through a 0.45 μm Millipore filter prior to analysis [30]. TEM was used to explore the morphology of the QEO-loaded micelles and blank micelles according to Da Silva [35]. The micelles were adsorbed on TEM grids by immersing the TEM grids into a droplet of the micellar solutions on a piece of Parafilm^®^ M, followed by air-drying of the grid for 24 h.

### 3.3. Preparation and Characterization of EO Nanohydrogels

QEO nanohydrogels were prepared through the protocol as follows: PF108 aqueous solution (1 mL) at a concentration of 20% (*w*/*v*) was prepared at 4 °C. QEO (100 μL) was added to the solution and mixed for 5 min at 25 °C by rigorous stirring. The resultant mixture was then subjected to a water bath of 37 °C for 10 min to obtain the QEO-loaded hydrogels, which were composed of self-crosslinked QEO-loaded PF108 micelles and also considered as a nanohydrogel formulation of QEO, i.e., QEO nanohydrogels. It is noted that the PF108 concentration of 20% (*w*/*v*) was selected based on a rheological study. Blank PF108 hydrogels with different polymer concentrations, i.e., 15%, 20%, and 25% (*w*/*v*) in deionized water, were prepared through the similar protocol as the micelle preparation. Rheological assessment was performed to measure the thermo-responsive properties of the hydrogels on a Kinexus Pro+ rheometer (Malvern Instruments, Bohlin Gemini HR Nano, UK) [36]. The elastic modulus (G’) and viscous modulus (G”) were investigated using parallel plates (20 mm diameter and 0.5 mm gap) in the oscillatory mode, with a temperature sweep from 20 to 38 °C at a heating rate of 3 °C/min. A constant oscillatory frequency (1 rad/sec) was applied in the temperature sweep test.

FTIR spectra of the freeze-dried QEO nanohydrogels and blank PF108 hydrogels were recorded using KBr pellets at room temperature on a Bruker Equinox 55 spectrometer (Equinox 55 Bruker Banner Lane, Coventry, Germany). Each spectrum was recorded in the range of 4000–400 cm^−1^ after the sample was scanned 16 times [37]. The samples were also investigated by XRD (PANalytical B.V., Netherlands) according to Munhuweyi [38]. The powder samples were placed on the sample holder and scanned in 0.02° steps from 5° to 50° (in 2θ) with 0.3 s per step.

### 3.4. Encapsulation Efficiency and Loading Capacity of QEO Nanohydrogels 

The QEO content in the QEO nanohydrogels was determined by UV–vis spectrophotometry (Genesys-20, Thermo-Scientific, Madison, WI, USA). QEO was extracted with n-hexane from the QEO nanohydrogels after the hydrogels were crushed in a high-speed homogenizer. The encapsulation efficiency (EE, %) and loading capacity (LC, %) of QEO were obtained based on the total weight of QEO used, the weight of QEO in the hydrogels and the weight of QEO nanohydrogels. The values of EE and LC were calculated according to Equations (1) and (2), respectively.
(1)$$\mathrm{EE}\;\left(\%\right)=\;\frac{\mathrm{Weight}\;\mathrm{of}\;\mathrm{QEO}\;\mathrm{in}\;\mathrm{hydrogels}}{\mathrm{Weight}\;\mathrm{of}\;\mathrm{total}\;\mathrm{QEO}\;}\;\;\times \;100\;$$
(2)$$\mathrm{LC}\;\left(\%\right)\;=\;\;\frac{\mathrm{Weight}\;\mathrm{of}\;\mathrm{QEO}\;\mathrm{in}\;\mathrm{hydrogels}}{\mathrm{Weight}\;\mathrm{of}\;\mathrm{QEO}\;\mathrm{hydrogels}}\;\times \;100\;$$

### 3.5. In Vitro QEO Release from QEO Nanohydrogels 

To simulate the QEO release from the QEO nanohydrogels on human skin as cosmetics, the weight loss of QEO in the nanohydrogels was monitored in the air as a function of time at 37 °C. The QEO nanohydrogels were placed on a plate in a thermostatic incubator (JINHONG, SHP-250, China) at 37 °C. At predetermined time intervals, the weight loss of QEO was obtained by weighing the QEO hydrogels. The pure QEO was used as a control in this study.

### 3.6. Antibacterial Activities of QEO Nanohydrogels 

*S. aureus* and *E. coli*, as representative Gram-positive and Gram-negative bacteria, respectively, were selected as test bacteria to determine the antibacterial activities of QEO and the QEO nanohydrogels [39]. Briefly, bacterial suspensions (200 μL) adjusted to 10^5^ CFU/mL were uniformly coated onto the surface of agar plates. Wells of 10 mm in diameter were punched in the inoculated agar medium with a sterile Pasteur puncher. Subsequently, 200 µL of QEO in n-hexane at a concentration of 100 μg/mL and the QEO nanohydrogels containing a comparable amount of QEO were added to each well. n-Hexane and the blank PF108 hydrogels were used as negative controls. The agar plates were incubated at 37 °C. The antibacterial activities were evaluated by measuring the inhibition zone diameters (excluding the diameter of the well). Each experiment was carried out in triplicate.

### 3.7. Statistical Analysis

The format of arithmetic mean value ± standard deviation (SD) was used for the descriptive data. The quantitative experiments were performed at least in triplicate. A *t*-test was performed to determine differences between groups, where *p* < 0.05 was considered to have significant differences

## 4. Conclusions

An innovative EO formulation of Qiai as QEO nanohydrogels was developed through a dual mechanism of EO-loading in PF108 micelles and self-crosslink of micelles. The QEO-loaded PF108 micelles showed an irregular spherical morphology with the particle size ranging from 110 to 180 nm in diluted aqueous solutions. The EE% and LC% of QEO nanohydrogels were as high as 80.2% and 6.8%, respectively. QEO nanohydrogels showed a sustained QEO release profile in a simulated body surface environment. A cumulative amount of 95% QEO could be slowly released from the QEO nanohydrogels in 35 h. EO nanohydrogels displayed stronger antimicrobial activity against both *S. aureus* and *E. coli* than pure QEO over a longer period of time. Due to these attractive properties and the good biocompatibility, the QEO nanohydrogels could be used as antibacterial gel cosmetics. Moreover, the thermo-responsive property of the QEO nanohydrogels could potentially allow them to be used as an antibacterial gel at body temperature, and convert to liquid status at low temperatures for storage and transportation.

## Figures and Tables

**Figure 1 molecules-26-05771-f001:**
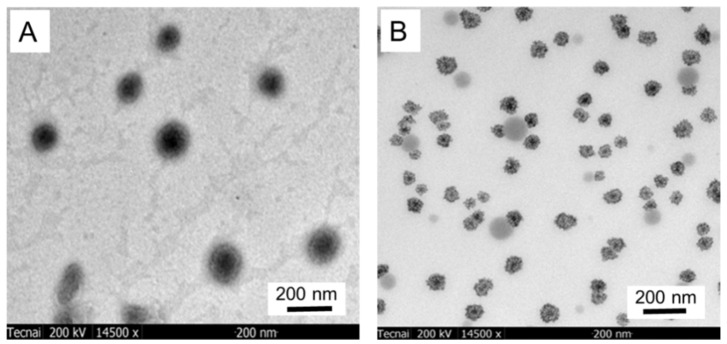
TEM analysis of the blank PF108 micelles (**A**) and QEO-loaded micelles (**B**).

**Figure 2 molecules-26-05771-f002:**
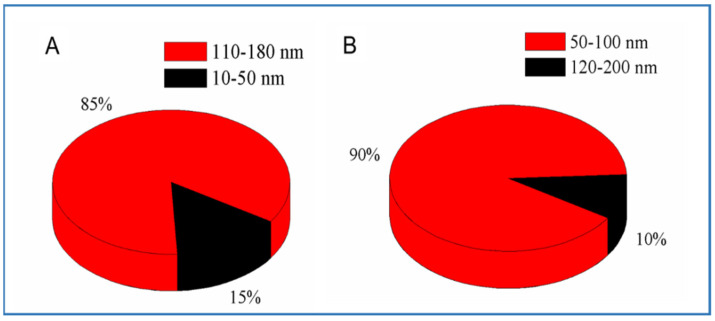
Particle sizes of the blank PF108 micelles (**A**) and QEO-loaded micelles (**B**) determined by DLS.

**Figure 3 molecules-26-05771-f003:**
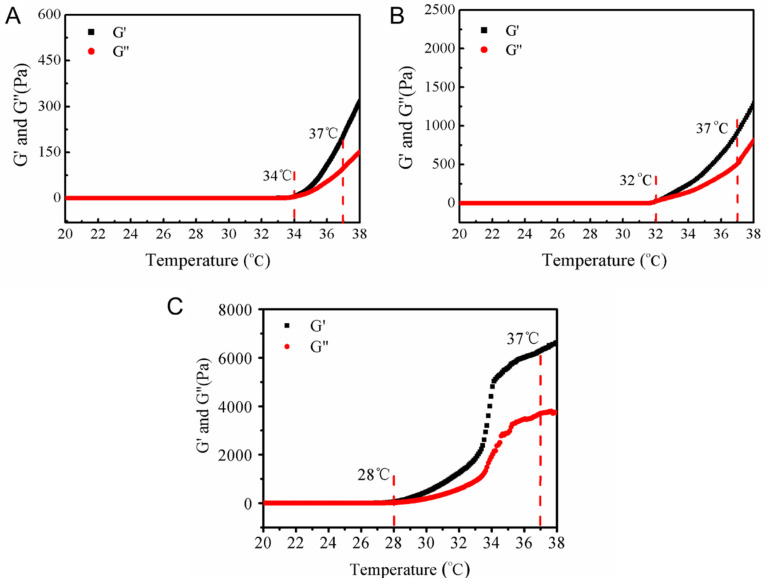
Rheological studies of PF108 hydrogels with different concentrations of PF108: (**A**) 15%, (**B**) 20%, and (**C**) 25% (*w*/*v*).

**Figure 4 molecules-26-05771-f004:**
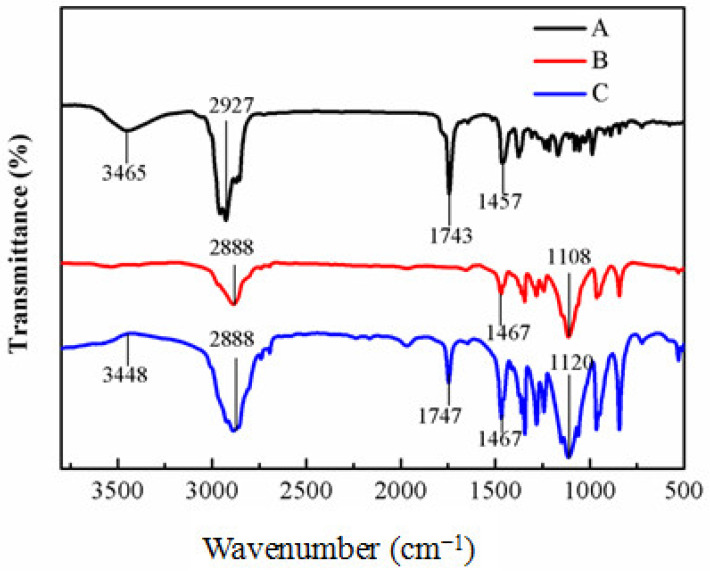
FTIR spectra of QEO (**A**), PF108 hydrogels (**B**), and QEO nanohydrogels (**C**).

**Figure 5 molecules-26-05771-f005:**
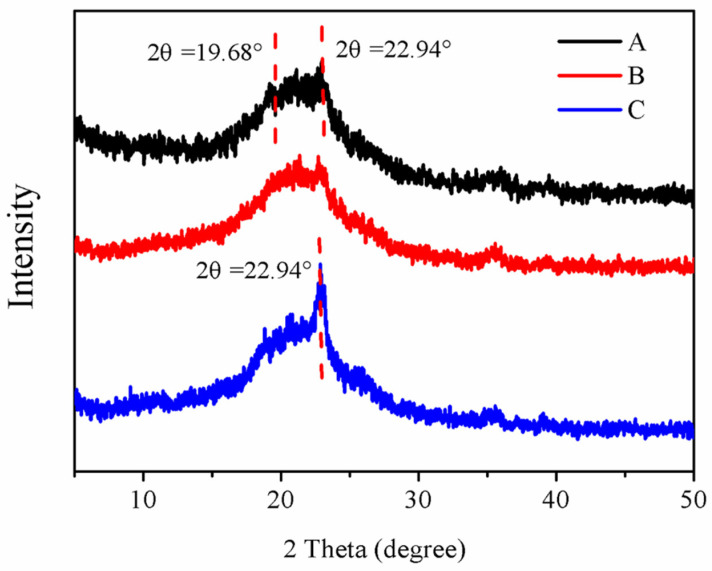
XRD spectra of PF108 powder (**A**), PF108 hydrogels (**B**), and QEO nanohydrogels (**C**).

**Figure 6 molecules-26-05771-f006:**
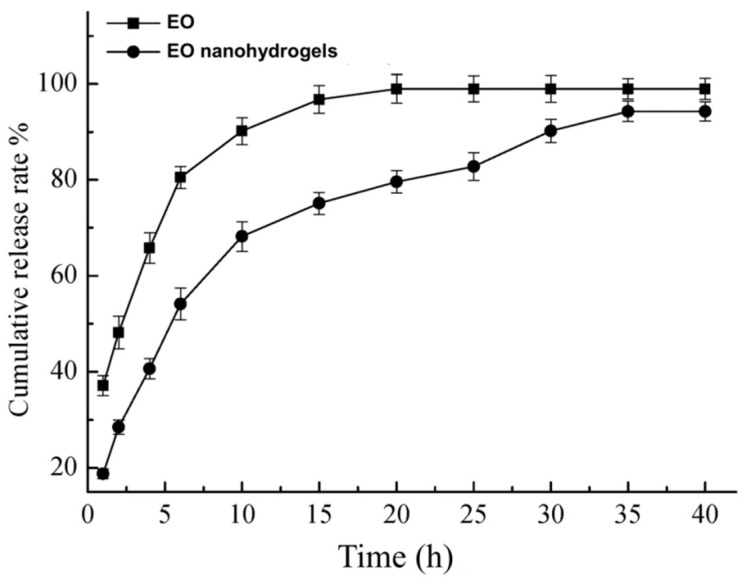
In vitro release of QEO nanohydrogels and free QEO.

**Figure 7 molecules-26-05771-f007:**
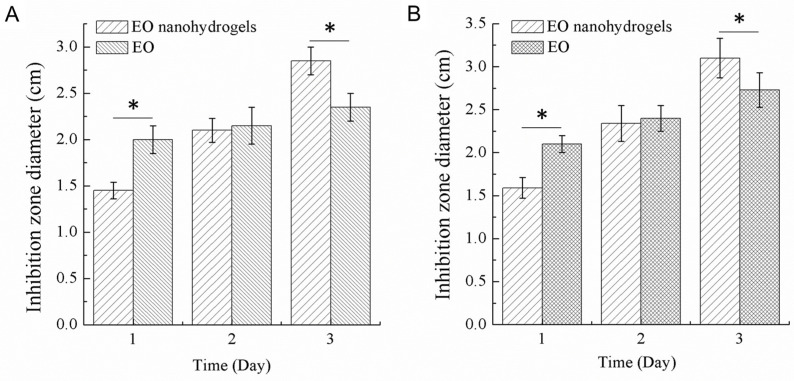
Antibacterial activities of QEO nanohydrogels and free QEO against *S. aureus* (**A**) and *E. coli* (**B**). An asterisk (*) indicates significant difference with *p* < 0.05.

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
