# Peer review of "Preparation and Antibacterial Activity of Thermo-Responsive Nanohydrogels from Qiai Essential Oil and Pluronic F108"

_molecules, 2021, doi:10.3390/molecules26195771_

Round 1

Reviewer 1 Report

The authors investigated formulation of essential oil as nanohydrogels using Pluronic F108 in aqueous solution. Having in mind the pharmacological importance of essential oil and at the same time high volatility, this topic deserves to be investigated.

However, the concept of the manuscript is very common and results are predictive. Since essential oil was used, if it is available, please consider providing the chemical composition of oil (GC/MS analysis).  

My recommendation is to add into the Introduction section more information related to Qiai essential oil, such as other pharmacological activities and chemical composition. It is mentioned that this essential oil possesses antibacterial activity, but additional information regarding possible mechanisms of activity and main compounds should be added.

Additionally, discussion should be improved. For example, section 2.6. contains just the explanation of the results and one statement “similar results were reported previously…” My recommendation is to expand the discussion and use literature data, for example to explain possible mechanisms of antimicrobial activity.

Please provide details such as producer and model for thermostatic incubator (for in vitro releasing).

Author Response

Dear Editor and Reviewers,

Thank you for your time and constructive comments for our manuscript (#molecules-1358359). We have revised the entire manuscript carefully according to the reviewers’ comments. Please note that the red lines showed the revision in the revised manuscript.

Reviewer #1:

The authors investigated formulation of essential oil as nanohydrogels using Pluronic F108 in aqueous solution. Having in mind the pharmacological importance of essential oil and at the same time high volatility, this topic deserves to be investigated.

  1. However, the concept of the manuscript is very common and results are predictive. Since essential oil was used, if it is available, please consider providing the chemical composition of oil (GC/MS analysis).  My recommendation is to add into the Introduction section more information related to Qiai essential oil, such as other pharmacological activities and chemical composition. It is mentioned that this essential oil possesses antibacterial activity, but additional information regarding possible mechanisms of activity and main compounds should be added.

Response:  Thanks for the suggestion. We have added the chemical composition of Qiai essential oils (QEO) in line 63- 68 of the revised manuscript as follows: QEO was mainly composed of thujone (14.94%), globulol (11.6%), 1,8-cineole (11.31%), 1-caryophyllene (8.11%), camphor (6.87%), α-terpineol,(4.69%), caryophyllene oxide (4.05%), borneol (3.95%), eugenol (3.01%), and germacrene D (2.81%) according to our measurement with gas chromatography-mass spectrometry. The antimicrobial mechanism of QEO has been found to be involved in simultaneous disruption of cell structures, such as the cytomembrane, which led to the leakage of essential molecules, such as protein and K+, resulting in cell death [Reference 20].

  1. Additionally, discussion should be improved. For example, section 2.6. contains just the explanation of the results and one statement “similar results were reported previously…” My recommendation is to expand the discussion and use literature data, for example to explain possible mechanisms of antimicrobial activity.

Response:  Thanks. In line 180-184, we have added: “The mycelial growth of A. niger could not be completely inhibited by free clove essential oil (CEO) at concentration as high as 3 mg/mL, while CEO encapsulated by chitosan nanoparticles could completely inhibit the fungal growth at 1.5 mg/mL [32]. Lemon myrtle essential oil (LM-EO) showed enhanced antibacterial activity against S.aureus, L. monocytogenes, and E.coli compared to LMEO alone [33].”

  1. Please provide details such as producer and model for thermostatic incubator (for in vitroreleasing).

Response:  Thank you. Producers and models of the Malvern Instrument and the  thermostatic incubator have been provided in line 223-224, and 247-248, respectively.

Reviewer 2 Report

The research topics are " Preparation and antibacterial activity of thermo-responsive nanohydrogels from qiai essential oil and pluronic F108", which is interest research, this manuscript demonstrated the use report an innovative formulation of EO as nanohydrogels, which were prepared through co-assembly of Qiai EO (QEO) and Pluronic F108 (PEG-b-PPG-b-PEG, or PF108) in aqueous solution, and the encapsulation efficiency and loading capacity of QEO reached 80.2% and 6.8%, respectively. This research also demonstrated QEO nanohydrogels showed stronger antimicrobial activity against S. aureus and E. coli than the free QEO due to the enhanced stability and sustained-release characteristics the research very interests, but some information in the manuscript is not clear. It is recommended major revision.
1.    Essential oil as aromatic or volatile oils, how to check the concentration after produce nanohydrogels?
2.    After storage, does the essential oil in nanohydrogels while decreasing?
3.    Figure 1 A resolution should be Improved.
4.    The figure 2, please provide the particle size profile information, it’s easier to know the nanohydrogels particle size. 
5.    Figure 7 shows the antibacterial activities of QEO and QEO nanohydrogels, and also demonstrated the sustained-release properties of QEO nanohydrogels, but there is no positive control such as antibiotic, please provide.
6.    General use of essential oil to produce Nanoemulsion solution or Nanoization would improve the antibacterial activity, but this manuscript research shows only expression sustained-release properties, why?

Author Response

Dear Editor and Reviewers,

Thank you for your time and constructive comments for our manuscript (#molecules-1358359). We have revised the entire manuscript carefully according to the reviewers’ comments. Please note that the red lines showed the revision in the revised manuscript.

Reviewer #2:

The research topics are " Preparation and antibacterial activity of thermo-responsive nanohydrogels from qiai essential oil and pluronic F108", which is interest research, this manuscript demonstrated the use report an innovative formulation of EO as nanohydrogels, which were prepared through co-assembly of Qiai EO (QEO) and Pluronic F108 (PEG-b-PPG-b-PEG, or PF108) in aqueous solution, and the encapsulation efficiency and loading capacity of QEO reached 80.2% and 6.8%, respectively. This research also demonstrated QEO nanohydrogels showed stronger antimicrobial activity against S. aureus and E. coli than the free QEO due to the enhanced stability and sustained-release characteristics the research very interests, but some information in the manuscript is not clear. It is recommended major revision.
1.  Essential oil as aromatic or volatile oils, how to check the concentration after produce nanohydrogels?

Response: Encapsulation efficiency and loading capacity rather than concentration were used to determined  content of EO in nanohydrogels. Encapsulation efficiency and loading capacity were were specifically  investigated in section 3.4.

  1.  After storage, does the essential oil in nanohydrogels while decreasing?

Fig.S  QEO nanohydrogels at different temperatures

Response: As shown in Fig. S, the hydrogels were in liquid state at 25 oC and in solid state  at 37 oC.  This study focused on exploring the potential application of the thermo-responsive property of the QEO nanohydrogels at the condition of body temperature. Therefore, the release at 37 oC is specially studied in section 3.5.  The release processes at other temperatures from 25 oC  to 37 oC are complicated and investigated in this study.
3.  Figure 1 A resolution should be Improved.

Response: We provided a new version of Figure 1 and the resolution has been improved.
4.  The figure 2, please provide the particle size profile information, it’s easier to know the nanohydrogels particle size. 

Response: Dynamic light scattering (DLS) is used to describe the particle size distribution in solution or suspension (Fig. 2). The morphology of ions cannot be observed by DLS. Thus, transmission electron microscopy was used to describe the specific size of nanoparticles in Fig. 1.

  1.  Figure 7 shows the antibacterial activities of QEO and QEO nanohydrogels, and also demonstrated the sustained-release properties of QEO nanohydrogels, but there is no positive control such as antibiotic, please provide.

Response: we have focused on the comparison the antibacterial activity of free essential oil and the encapsulated essential oil (QEO). The results showed that essential oil encapsulated by PF108 improved the antibacterial activity. In our previous study, we have specially investigated the antibacterial activity of Qiai essential oil and  positive drugs, ampicillin sodium and streptomycin sulfate in literature [20]. Therefore, we did not add additional positive controls in Figure 7.
6. General use of essential oil to produce Nanoemulsion solution or Nanoization would improve the antibacterial activity, but this manuscript research shows only expression sustained-release properties, why?

Response: Antibacterial activity of Qiai essential oil (QEO) is limited due to their poor water solubility and volatility. Some literatures have reported that the thermo-responsive property of PF108 is attributed to the PPG block, which is water-soluble at low temperatures while becomes insoluble at higher temperatures. Qiai essential oil is encapsulated by PF to form water soluble micelles, which can be observed directly during the experiment. Therefore, the authors focused on the effect of sustained-release properties on antibacterial activity.

Thanks for your time and assistance.

Round 2

Reviewer 1 Report

The authors revised and corrected the Manuscript according to most of the suggestions. Therefore, I recommend accepting of the manuscript.

Reviewer 2 Report

Paper have improved after recommendations